# Superlubric polycrystalline graphene interfaces

Xiang Gao[1], Wengen Ouyang [2], Michael Urbakh [1✉] & Oded Hod [1]

The effects of corrugated grain boundaries on the frictional properties of extended planar graphitic contacts incorporating a polycrystalline surface are investigated via molecular dynamics simulations. The kinetic friction is found to be dominated by shear induced buckling and unbuckling of corrugated grain boundary dislocations, leading to a nonmonotonic behavior of the friction with normal load and temperature. The underlying mechanism involves two effects, where an increase of dislocation buckling probability competes with a decrease of the dissipated energy per buckling event. These effects are well captured by a phenomenological two-state model, that allows for characterizing the tribological properties of any large-scale polycrystalline layered interface, while circumventing the need for demanding atomistic simulations. The resulting negative differential friction coefficients obtained in the high-load regime can reduce the expected linear scaling of grain-boundary friction with surface area and restore structural superlubricity at increasing length-scales.

[1] Department of Physical Chemistry, School of Chemistry, The Raymond and Beverly Sackler Faculty of Exact Sciences and The Sackler Center for Computational Molecular and Materials Science, Tel Aviv University, Tel Aviv, Israel. [2] Department of Engineering Mechanics, School of Civil Engineering, Wuhan University, Wuhan, Hubei, China. ✉email: urbakh@tauex.tau.ac.il

R eduction of energy dissipation and wear is of critical importance and growing demand in a wide range of areas, including mechanical, electronic, and biological systems. Compared to traditional lubrication schemes, structural superlubricity, featured as extremely low friction due to the effective cancellation of lateral forces at incommensurate crystalline interfaces, has emerged as a novel route toward efficient reduction of friction and wear[1,2]. Recent studies have reported structural superlubricity in nano- and micro-sized monocrystalline samples of layered materials, where misoriented lattices[3–7] or mismatched lattice constants[8–13] facilitate the necessary incommensurability. However, scaling up structural superlubricity towards macroscopic dimensions inevitably involves forming junctions between surfaces of polycrystalline nature.

The simplest example of polycrystalline monoatomic two-dimensional (2D) surface is polycrystalline graphene (PolyGr), which is composed of randomly shaped and oriented single crystalline graphene patches separated by sharp grain boundaries (GBs). The latter are characterized by chains of lattice dislocations often including pentagon–heptagon pairs[14]. Since the friction between misaligned graphitic patches is known to be negligibly small[3], one could expect that superlubricity would prevail also in large-scale PolyGr interfaces, pending that the patch dimensions remain at the nanoscale. However, PolyGr GBs often exhibit out-of-plane corrugation[15–17], which may introduce substantial friction[18–20] and enhance wear[21]. To control and eliminate such undesired effects one must fully understand the mechanisms underlying energy dissipation at elongated graphene GBs. This, in turn, requires elucidating collective effects between different topological defects along the GBs under persistent shear.

Here, we reveal that collective dynamic effects at PolyGr GBs may lead to unusual nonmonotonic variation of the friction with normal load and temperature. Notably, at room temperature or above we find a monotonic decrease of friction with the external normal load for the systems considered, resulting in negative differential friction coefficients. The discovered phenomenon is of general nature and is expected to occur in other polycrystalline layered materials that demonstrate out-of-plane GB deformations. Moreover, the knowledge gained in this study may provide insights regarding universal mechanisms of energy dissipation appearing in extended multi-contact rough interfaces, where the formation and rupture of contacts dictates the friction[22–27].

## Results

### Simulation setup
Our model system for studying the friction over extended graphene GBs is shown in Fig. 1a. From top to bottom, the system consists of a slider composed of three Bernal stacked pristine graphene (PrisGr) layers oriented at $\theta_0 = 38.2°$, and a substrate consisting of a layer of PolyGr including two patches with orientation angles of $\theta_1 = 0°$ (left and right parts of the supercell, see Fig. 1b) and $\theta_2 = 8°$ (middle section of the supercell, see Fig. 1b), and two Bernal stacked PrisGr layers oriented at $\theta_3 = 0°$, where the bottom one ($l_6$) is kept fixed. All orientation angles are measured between the sliding direction (x-axis) and the armchair direction of the lattice of the relevant layer. The PolyGr layer contains two GBs composed of lines of separated pentagon–heptagon pair dislocations along the GB (y-axis) direction (Fig. 1b). The dislocations introduce in-plane strain to the otherwise hexagonal lattice, which is partially relieved via out-of-plane deformations[18]. The average corrugation of the free surface (bottom three layers without the slider) reaches ~1.4 Å, consistent with previous experimental measurements[16]. After annealing the entire six layers structure at 1000 K and zero normal load, most of the dislocations exhibit considerable out-of-plane deformations, protruding upward or downward as shown in the lower panel of Fig. 1b. Imposing normal load, by adding a uniform force in the vertical direction to each atom in the top layer, reduces the average corrugation of the dislocations from 0.5 Å to 0.1 Å as the load increases from 0 to 1.9 GPa (Fig. 1c).

In the sliding simulations, normal loads of up to ~2.3 GPa are applied. The top layer is kept rigid and is shifted with a constant velocity of $v_0 = 5\,m/s$ in the x direction (Fig. 1d). Since the slider layers are incommensurate with both PolyGr patches, the shear plane appears at the interface of the third layer of the slider ($l_3$) and the PolyGr layer ($l_4$). The generated heat is removed from both slider and substrate without affecting the dynamics at the sheared interface. See "Methods" and Supplementary Methods for further details regarding the simulation model and protocols.

### Load and temperature dependence of friction
The dependence of the interfacial friction between the PrisGr and PolyGr surfaces on the normal load and temperature are presented in Fig. 2a, b, respectively. Figure 2a presents the load dependence of the friction for several temperatures in the range 0–300 K. At zero temperature the friction shows nonmonotonic behavior with load, where an expected friction increase with load is observed up to ~0.5 GPa followed by friction reduction at higher loads. The corresponding effective friction coefficients obtained at the low- and high-pressure regimes are $1.5 \times 10^{-4}$ and $-8.1 \times 10^{-5}$, respectively, well within the superlubric regime. When the temperature is increased to 50 K the overall friction increases and the maximal friction point shifts towards lower normal loads. When the temperature is further increased to 150 K the maximum completely disappears and the friction reduces monotonically with the normal load exhibiting negative differential friction coefficients across the entire load range considered[13,28]. At 300 K a similar behavior is observed with overall reduced friction. The differential friction coefficients calculated in this case are between $-1 \times 10^{-4}$ and $-8.9 \times 10^{-6}$. Compared to experiments, the maximal frictional stress obtained in our simulations (~160 kPa) is less than one order magnitude higher than that measured for misaligned homogeneous pristine graphitic contacts[4,29], and comparable to that of aligned graphite/hexagonal boron nitride (h-BN) heterojunction[10]. See Supplementary Note 1 for further details regarding the estimation of friction coefficients.

The friction–temperature relation, shown in Fig. 2b, exhibits a similar behavior. At zero normal load, the friction presents a nonmonotonic curve with a maximal value at $T = 150\,K$. As the normal load increases, the overall friction force decreases, and the position of maximal friction shifts to lower temperatures.

### Mechanisms of energy dissipation
The nonmonotonic behavior in the friction–load and friction–temperature relations suggests that there are two competing effects that dictate the overall energy dissipation throughout the sliding process. To understand the underlying mechanisms, we examine the simple case of friction–load relation in the absence of thermal effects, i.e. at $T = 0\,K$. Analyzing the energy dissipation routes by calculating the steady-state dissipation power through the damped layers in different directions (Fig. 3a) demonstrates that energy dissipation is dominated by out-of-plane atomic motion at the GBs regions (see Fig. 3a, b and Supplementary Note 2). Based on this understanding we tracked the vertical motion of the atoms with maximal root-mean-square (RMS) corrugation in each dislocation during sliding (Fig. 3c). The RMS corrugation is calculated by temporal averaging the out-of-plane displacements of each atom within a given dislocation over the entire trajectory.

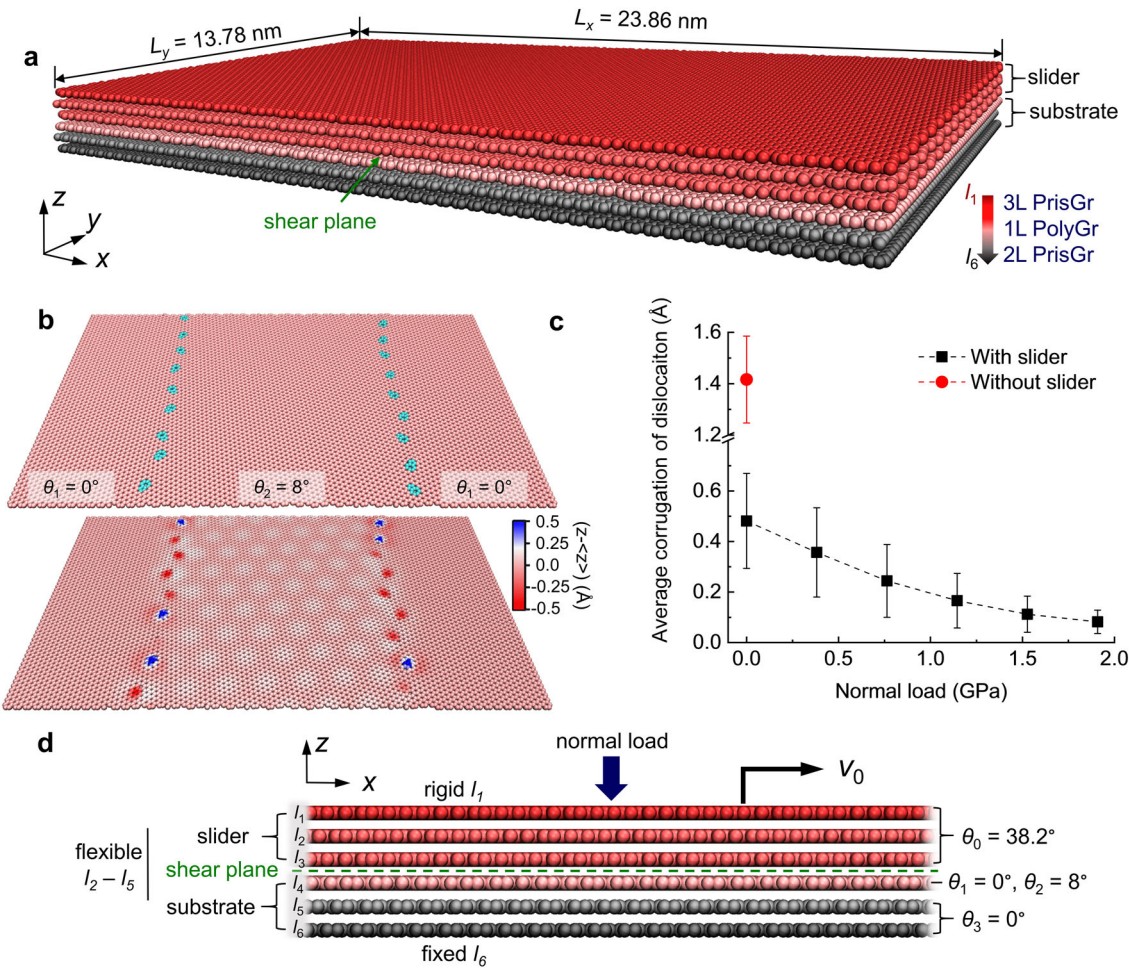

**Fig. 1 Molecular dynamics simulation model and setup. a** Perspective view of the model system. From top to bottom, the system contains three layers of Bernal stacked pristine graphene (red spheres) with orientation angle $\theta_0 = 38.2°$, one layer of polycrystalline graphene (pink and cyan spheres) with orientation angles $\theta_1 = 0°$ and $\theta_2 = 8°$ for the two patches in the layer, and two layers of pristine graphene (grey spheres) with orientation angle $\theta_3 = 0°$. The dark red and grey colored spheres indicate the rigid top layer and the fixed bottom layer atoms, respectively. Periodic boundary conditions are applied along both lateral directions. **b** Atomic structure and annealed topography of the polycrystalline graphene layer within the stack. Pink and cyan spheres represent carbon atoms locally associated with pristine graphene or GB heptagon–pentagon defects. The lower panel shows the surface corrugation (see corresponding color bar) at zero normal load with respect to the average height, $\langle z \rangle$, of the two grains. **c** The average dislocation corrugation as a function of normal load (black squares). As reference, we present the corresponding corrugation calculated in the absence of the slider (red circle). The error bars present standard deviations, obtained by averaging over all the dislocations. **d** Sliding simulation setup. The shear plane at the interface between layers $l_3$ and $l_4$ is denoted by the green line (see also green arrow in panel (**a**)). During the simulations, the top rigid layer is shifted at a constant velocity $v_0 = 5\,\text{m/s}$ along the $x$-axis direction (the armchair direction of the pristine graphene substrate layers). Normal load is applied to the top layer by adding a uniform force to each of the atoms. The bottom layer is kept rigid and fixed in place throughout the motion. The dynamics of all other atoms follows the REBO intralayer potential and the registry-dependent interlayer potential. Atoms in layers $l_2$ and $l_5$ are further subjected to damped dynamics at zero temperature or Langevin thermostats at finite temperatures, as detailed in the "Methods" section and in Supplementary Methods.

Interestingly, we observe that at zero and 0.6 GPa normal loads several dislocations undergo dynamic snap-through buckling between an upward protrusion state and a downward protrusion state (see top and middle panels in Fig. 3c and Supplementary Movies 1 and 2), which resembles the snap-through dynamics of an arched beam. These dynamically buckling dislocations correspond to the high energy dissipation sites shown in Fig. 3b and Supplementary Fig. 8. Notably, for dislocations that do not buckle during sliding, the associated energy dissipation is very low, comparable to that in the grain areas. This demonstrates that the surface corrugation of the GB alone does not produce significant dissipation and the snap-through buckling of dislocations provides the major contribution to the enhanced friction exhibited by GBs and to the corresponding nonmonotonic frictional dissipation behavior with normal load.

The dynamic snap-through buckling phenomenon indicates that the GB dislocations exhibit bi-stable behavior characterized by the upward and downward protruding states separated by transition energy barriers (TEBs). An estimate of the corresponding barrier heights can be obtained by considering the thermally activated dislocation buckling process in the absence of sliding. To this end, we performed room temperature ($T = 300\,\text{K}$) equilibrium molecular dynamic simulations of the six-layered model system presented in Fig. 1, without applying any shear force. During the dynamics, we followed the $z$ coordinate of the atom with largest RMS corrugation in the dislocation region. A typical example of such an atomic trajectory is presented in the inset of Fig. 4a showing clear evidence of a mechanical bistability. The buckling Helmholtz free-energy, $A(\Delta z)$, profile can be extracted from such trajectories by evaluating the probability density distribution, $\rho(\Delta z)$,

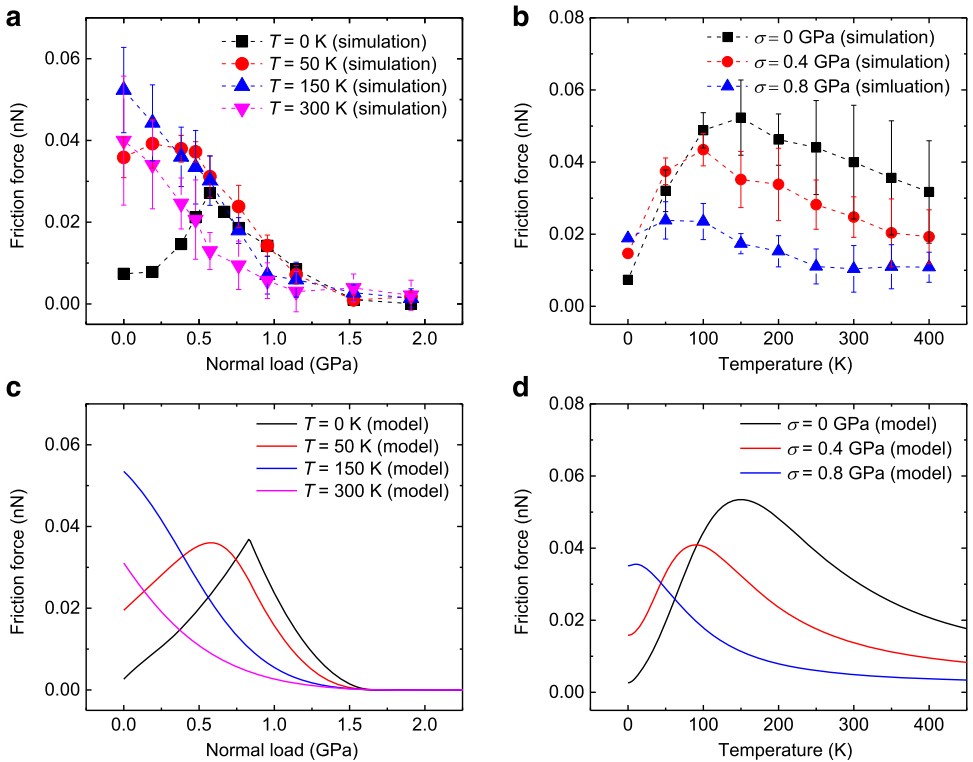

**Fig. 2 Load and temperature dependence of friction.** Molecular dynamics simulation results of **a** load dependence of friction at representative temperatures of $T = 0$, 50, 150, and 300 K and **b** temperature dependence of friction at representative normal loads of $\sigma = 0$, 0.4, and 0.8 GPa for the model system presented in Fig. 1. Here, the supercell contains a total grain boundary of length $\sim 28$nm. **c**, **d** show results obtained using the phenomenological model for **c** the load dependence of friction at several temperatures, and **d** the temperature dependence of friction under several normal loads. The model parameters extracted from the dynamical simulations, are: $\triangle E_1 = 0.01$ eV, $\triangle E_2 = 0.1$ eV, $\alpha_1 = 0.06$ eV/GPa, $\alpha_2 = 0.03$ eV/GPa, $\beta = 0.3$, $c_0 = 0.005$ eV, $\Delta x = 10.8$ Å, $f_0 = 1.5 \times 10^{11}$ s$^{-1}$, $N = 18$, and $v_0 = 5$ m/s. The error bars in panels **a** and **b** represent the standard deviations obtained from averaging friction forces of 3–13 consecutive sliding periods. Representative lateral force traces at different normal loads and temperatures, and further details regarding the friction force calculations can be found in Supplementary Note 1. Additional details regarding the phenomenological model are given in Supplementary Note 6.

of finding this atom at a displacement of $\Delta z$ away from the average height of the two PolyGr layer grains, and substituting it in the relation $A(\Delta z) = -k_B T \ln \left[ \rho(\Delta z) \right]$, where $k_B$ is the Boltzmann constant. From this profile the snap-through buckling process barrier heights can be readily extracted (see e.g. $\triangle E_b$ and $\triangle E_u$ in Fig. 4a). We note that the TEBs obtained from this method compare well with those obtained using nudged elastic band (NEB) calculations (see Supplementary Note 4). The corresponding distribution of dislocation barriers appears in Fig. 4b. Notably, by performing the same analysis under an external normal load we find that, along with the reduction of the spatial corrugation of the dislocations (Fig. 1c), the average TEB between the upward and downward protruding states and the number of dislocations with substantial barrier reduce as well (Fig. 4c). Furthermore, we find a positive correlation between the TEB for (un)buckling of a given GB dislocation and its out-of-plane corrugation (Fig. 4d). Importantly, the energy profile along the vertical dislocation buckling trajectory varies with the lateral displacement of the slider (see Fig. S13 of Supplementary Note 4). These variations stem from the fact that, by construction, the adopted classical interlayer potential (ILP, see "Methods" section) accounts for the interlayer Pauli repulsions between electronic clouds associate with atoms residing on adjacent layers, which are registry dependent. Therefore, at certain positions (e.g. the purple line in Fig. S13b) the energy barrier along the buckling trajectory vanishes allowing for buckling to occur even at zero temperature. Further variations of the buckling energy profile

during sliding may result in reverse buckling, thus manifesting dynamic dislocation buckling during sliding.

This analysis allows us to identify the competing effects leading to the nonmonotonic friction dependence on normal load appearing in Fig. 2a. At zero temperature ($T = 0$ K) the snap-through bi-stability can be triggered by the sliding process via two main effects: (i) variation of the snap-through barrier along the scan-line and (ii) shear-force induced out-of-plane motion of atoms in the dislocation region. The former results from the registry dependence of the vertical buckling energy profile, as discussed above, whereas the latter, which may be caused by local heating, is found to be of minor importance under our simulation conditions, as demonstrated below by comparison to a phenomenological two-state model. At zero normal load, only a few shear-induced buckling events occur (see top panel of Fig. 3c and Supplementary Movie 1) due to the relatively high-energy barriers with respect to the kinetic energy of the dislocation atoms. However, once triggered, each snap-through event generates a large kinetic energy pulse (see top panel of Fig. 3d) that dissipates through inter- and intralayer interactions. Increasing the normal load to 0.6 GPa results in a reduction of the TEBs and hence an increase in the fraction of buckling dislocations (see middle panel of Fig. 3c and Supplementary Movie 2). This, in turn, leads to more frequent kinetic energy pulses (see middle panel of Fig. 3d), consistent with the observed enhancement in the out-of-plane energy dissipation and friction

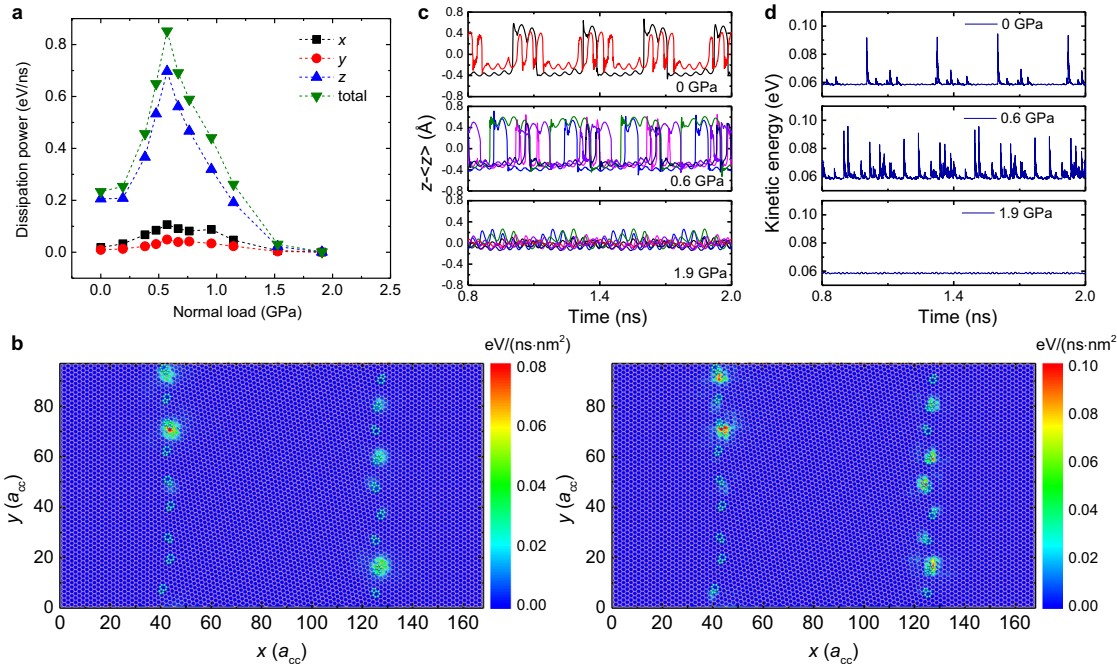

**Fig. 3 Mechanism of the load dependence of frictional dissipation at zero temperature. a** Total dissipation power and its directional contributions ($x, y, z$ —see Fig. 1a) as a function of normal load. **b** Spatial distributions of dissipation power density in the $z$ direction of the second pristine layer (left panel) and the fifth pristine layer (right panel) under a normal load of 0.6 GPa. The geometric configuration of the PolyGr layer is superimposed on the 2D maps. The pentagon–heptagon atoms are shown in cyan and the hexagon carbon atoms are represented in pink. The power density is calculated based on area elements with size of $a_{cc}^2$ ($a_{cc} = 1.42039$ Å is the equilibrium C–C bond length in the REBO potential). **c** Representative dislocation trajectories (for two sliding periods at steady state) corresponding to the out-of-plane motion of the atoms with maximal RMS corrugation within each dislocation (different colors correspond to different dislocations) during sliding, under normal loads of 0, 0.6, and 1.9 GPa (from top to bottom panels, respectively). Substantial buckling and unbuckling dynamics during sliding is observed under normal loads of 0 and 0.6 GPa. **d** Kinetic energy profiles corresponding to the traces appearing in panel (**c**). More details regarding the temperature effects on dislocation buckling can be found in Supplementary Note 3. Movies of typical simulations are also provided in the Supplementary Information.

(upward facing blue triangles in Fig. 3a). When the normal load is further increased to 1.9 GPa, the barrier height becomes comparable to the kinetic energy of the dislocation atoms and the transition between upward and downward protruding states occurs smoothly (showing no abrupt kinetic energy pulses, see bottom panel of Fig. 3d) with lower amplitude and little energy dissipation (see bottom panel of Fig. 3c and Supplementary Movie 3). This results in reduced out-of-plane dissipation and friction as seen in Fig. 3a at this load. A similar behavior is obtained at a higher temperature of $T = 50$ K, however, with a downshift of the friction maximum due to thermal activation of the buckling process (see Fig. 2a and Supplementary Movie 4). At even higher temperatures of $T = 150$ and $300$ K, most of the dislocations buckle already at zero normal load and the friction force decreases monotonically with increasing load due to the reduction of the TEBs (see Fig. 2a and Supplementary Movies 5 and 6).

Next, consider the nonmonotonic friction dependence on temperature (Fig. 2b). At low temperatures ($T < 150$ K) and in the absence of normal load, thermal fluctuations can assist overcoming the TEBs for buckling during sliding, activating more dislocations to buckle and increasing their snap-through frequency (see Supplementary Note 3 and Supplementary Movie 4). Since each such event is accompanied by a sharp kinetic energy dissipation pulse, the overall friction increases. As the temperature further increases, the thermal energy, $k_B T$, becomes comparable to the energy barrier heights, leading to frequent spontaneous buckling of dislocations between the two states (see Supplementary Note 3 and Supplementary Movie 6).

Correspondingly, less energy is invested by the sheared slider to induce dislocation buckling, and the friction reduces. Therefore, we conclude that the nonmonotonic friction–temperature relation stems from two competing effects, i.e. thermally assisted shear-induced buckling vs. thermally dominated spontaneous buckling. A similar behavior is obtained at higher normal loads; however, the corresponding reduced barriers result in lower overall friction and a downshift of the friction maximum (Fig. 2b).

**Robustness of the results**. To evaluate the general nature of this frictional behavior and the sensitivity of our results towards different simulation parameters, we performed additional simulations using thicker model systems (stacks of 8 and 10 layers). The results demonstrate similar dynamic snap-through buckling of GB dislocations and a corresponding nonmonotonic load dependence of the friction (see Supplementary Note 5). This indicates that the six-layered model system employed herein is adequate to reflect the frictional behaviors of thicker (and stiffer) systems. We have also repeated some of our simulations using different slider velocities of 2 and 10 m/s, demonstrating similar dynamic snap-through GB dislocation buckling, which leads to a nonmonotonic to monotonic transition of the friction–load dependence with increasing temperature (see Supplementary Note 5). For finite temperatures, we find that increasing the sliding velocity leads to a mild shift of the friction force peak position toward higher normal loads. This results from the fact that at higher velocities the timescale for each individual buckling

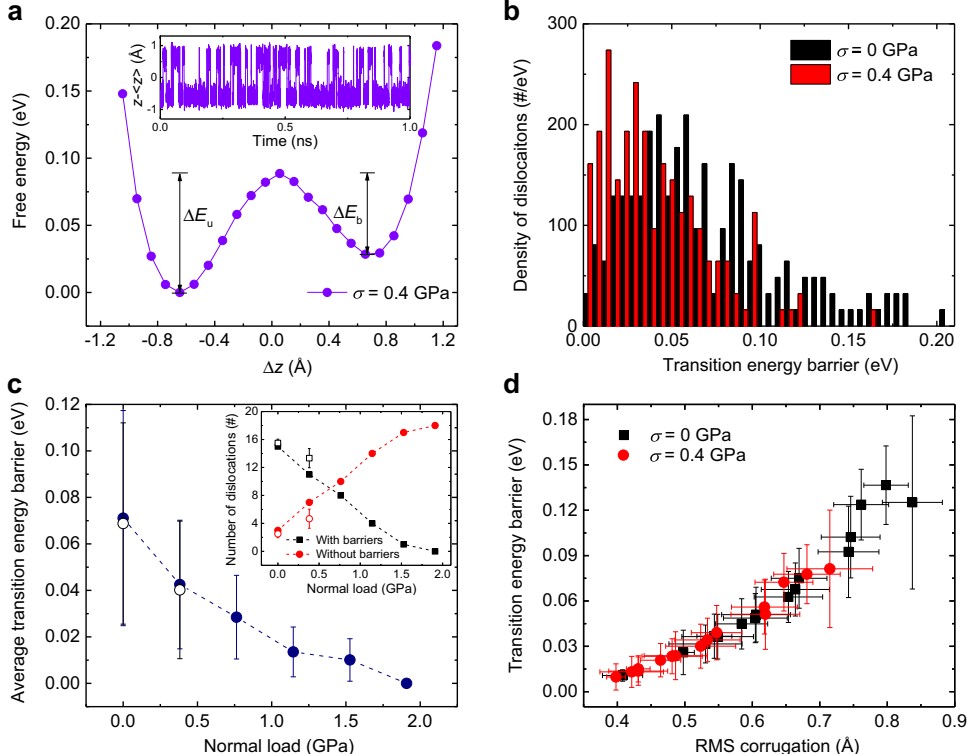

**Fig. 4 Dislocation buckling transition energy barriers. a** An example of a free energy profile of one dislocation calculated under a normal load of 0.4 GPa at the initial configuration (prior to sliding). The inset shows a 1 ns atomic out-of-plane motion trajectory of the dislocation during the equilibrium simulation. **b** The number density distribution of finite transition energy barrier dislocations (spatially averaged over six equidistant slider positions along one sliding period) calculated under normal loads of 0 (black bars) and 0.4 GPa (red bars). **c** The average transition energy barrier as a function of normal load. The average values (full symbols) and error bars (standard deviation of the distribution presented in panel (**b**) are calculated at the initial configuration over all dislocations of transition energy barriers above 0.0026 eV. The empty data points represent the spatial average results (over six interlayer positions within the sliding period), producing essentially the same results within the given error bars. The inset shows the number of dislocations with barriers smaller (full red circles) or higher (full black squares) than 0.0026 eV, as a function of normal load. The empty symbols present the spatially averaged results. **d** Correlation between dislocation corrugation and its transition energy barrier calculated under normal loads of 0 (black squares) and 0.4 GPa (red circles). The calculation is performed separately for each dislocation, spatially averaged over six equidistant slider positions along one sliding period as in panel **b**. The error bars represent the corresponding standard deviation. All calculations presented in this figure were performed at $T = 300$ K.

event is reduced, thus reducing the probability of thermal activation to promote buckling. In such case, a higher normal load is required for buckling to occur. The fact that we find only mild quantitative changes in the friction with sliding velocity suggests that the mechanism underlying the frictional behavior predicted by our simulations is insensitive to variations in the sliding velocity within the wide velocity range considered. In addition, we considered also a PolyGr surface of a smaller GB misfit angle of $\theta_2 = 2.5°$ that exhibits higher GB corrugation than the $\theta_2 = 8°$ case. We find that the overall qualitative dependence of the friction on the normal load and temperature is preserved with some quantitative modifications due to the increased barrier heights (see Fig. 2a and Supplementary Note 5).

**Load–temperature duality**. We note that the fact that we obtain similar nonmonotonic behaviors of the friction with normal load and with temperature, as presented in Fig. 2a, b, is not coincidental. In the load-dependence case, the thermal energy remains constant, and the barrier heights are varied relative to it upon changing the external load, whereas in the temperature-dependence case the thermal energy is varied with respect to the barrier heights by controlling the target temperature of the thermostats. The overall net effect is therefore very similar.

**Two-state phenomenological model**. The dominance of the bistability mechanism on the frictional properties of the GBs allows us to provide a simple and intuitive description of the tribological properties of this involved system. This is achieved by mapping the complex out-of-plane buckling dynamics onto a simple two-state model, associated with the upward and downward protruding states of a single dislocation, which are separated by a TEB (see Fig. 4a), $\triangle E^n(\sigma)$, $n$ being the dislocation number. Based on our simulation results, we introduce the following assumptions: (i) the distribution of TEBs corresponding to different dislocations is uniform within a given range (Fig. 4b); (ii) the reduction of the TEB of any given dislocation with the normal load is approximately linear (Fig. 4c); and (iii) the TEB of all dislocations depend on the relative position, $x$, of the interfacing layers at the sliding contact, such that $\triangle E^n(\sigma) \rightarrow \triangle E^n(x, \sigma)$. The latter assumption results from the variation of the Pauli repulsions experienced by the two interfacing layers when their hexagonal lattices slide across each other, as discussed above. Note that $\triangle E_{max}^n(\sigma) \geq \triangle E^n(x, \sigma) \geq \triangle E_{min}^n(\sigma)$, where $\triangle E_{max}^n(\sigma)$ and $\triangle E_{min}^n(\sigma)$ are the maximal and minimal buckling TEB heights encountered along one period of the sliding path, set by the hexagonal lattice periodicity of the pristine lower surface of the slider.

Considering first the zero-temperature case, for a given normal load, $\sigma$, buckling will occur only in dislocations for which the TEB

vanishes along the sliding path, $\triangle E_{\min}^n(\sigma) = 0$. The energy dissipated once buckling occurs equals the energy invested to induce the buckling, which in turn, is given by the maximal TEB height along the sliding path, $\triangle E_{\max}^n(\sigma)$. As stated above, the TEBs are assumed to reduce linearly with the normal load such that:

$$\begin{cases} \triangle E_{\min}^n(\sigma) = \triangle E_{\min}^n(0) - \alpha_1\sigma \\ \triangle E_{\max}^n(\sigma) = \triangle E_{\max}^n(0) - \alpha_2\sigma \end{cases}. \quad (1)$$

The slopes, $\alpha_i$, and the constants $\triangle E_{\min/\max}^n(0)$ can be estimated from our simulation results (Fig. 4c). The average load-dependent friction force can then be estimated as follows:

$$F_f(\sigma) = \sum_n^N \frac{\triangle E_{\max}^n(\sigma)}{\triangle x}\left[1 - H\big(\triangle E_{\min}^n(\sigma)\big)\right]H\big(\triangle E_{\max}^n(\sigma)\big). \quad (2)$$

Here, $\triangle E_{\max}^n(\sigma)/\triangle x$ is the average friction force induced by the buckling of the $n$th dislocation, calculated as the dissipated energy per period. The periodicity is determined by the orientation of the pristine lower slider layer relative to the sliding direction. In the present case, the slider is rotated by 38.2° with respect to the sliding direction (the armchair axis of the pristine substrate layers). The translation vector along this direction obeys the relation $\mathbf{T} = 2\mathbf{a}_1 + 11\mathbf{a}_2$ (see Fig. S14), where $\mathbf{a}_1$ and $\mathbf{a}_2$ are the basis vectors of the pristine graphene lattice. The corresponding periodicity is $|\mathbf{T}| = \sim 3\,\mathrm{nm}$. The first Heaviside step function $H(x)$ in Eq. (2) takes into account that at zero temperature only dislocations of vanishing TEB buckle, the second Heaviside function assures that unphysical negative TEB values are not considered, and the sum runs over all $N$ dislocations.

Equation (2) demonstrates that the nonmonotonic friction dependence on external load originates from two competing effects: (i) the increase of number of active dislocations (dislocations with vanishing $\triangle E_{\min}^n(\sigma)$) with normal load and (ii) decrease of the dissipated energy per buckling event $\big(\triangle E_{\max}^n(\sigma)\big)$.

For simplicity, the sum appearing in Eq. (2) can be approximated via integration. To this end, we further assume that there is a proportionality relation between the maximal and minimal values of the TEB along a sliding period at zero normal load:

$$\triangle E_{\min}^n(0) = \beta\triangle E_{\max}^n(0) - c_0 \quad (3)$$

Together with Eq. (1) this yields the following relation between the maximal and minimal values of the TEB along a sliding period at any finite normal load, $\sigma$:

$$\triangle E_{\min}^n(\sigma) = \beta\triangle E_{\max}^n(0) - \alpha_1\sigma - c_0 = \beta\triangle E_{\max}^n(\sigma) - c_0 - (\alpha_1 - \alpha_2\beta)\sigma. \quad (4)$$

With this, we may approximate Eq. (2) as follows:

$$F_f(\sigma) \approx N\int\frac{\triangle E_{\max}(\sigma)}{\triangle x}\left[1 - H\big(\triangle E_{\min}(\sigma)\big)\right] \\ H\big(\triangle E_{\max}(\sigma)\big)P_b\big(\triangle E_{\max}(0)\big)\mathrm{d}\triangle E_{\max}(0), \quad (5)$$

where we introduced the probability density distribution of the TEB heights $P_b\big(\triangle E_{\max}(0)\big)$ (assumed to be normalized to 1) and used the fact that $\mathrm{d}\triangle E_{\max}(0) = \mathrm{d}\triangle E_{\max}(\sigma)$, per Eq. (1). Based on our simulation results (Fig. 4b), we may approximate the probability distribution $P_b\big(\triangle E_{\max}(0)\big)$ to vanish outside a finite range $\left[\triangle E_1, \triangle E_2\right]$ and be uniform within it. Using this in Eq. (5) and performing the integral results in the black line appearing in Fig. 2c, which agrees well, both qualitatively and quantitatively, with the simulation results. The fact that the phenomenological model captures well the atomistic simulation results without considering local heating effects indicates that the latter have

minor contribution to the friction force under the studied conditions.

At finite temperatures, buckling is not limited to zero barrier dislocations, and can occur also for finite barrier dislocations. To account for this, we introduce the survival probability function, $p(t, \sigma)$, that describes the probability of a given dislocation not to buckle up to time $t$, along one sliding period. Assuming that the dislocations are independent, this can be described as a first-order rate equation:

$$\frac{\mathrm{d}}{\mathrm{d}t}p(t, \sigma) = -f_0 e^{\frac{-\triangle E^n(t,\sigma)}{k_B T}}p(t, \sigma), \quad (6)$$

where $f_0$ is constant, $\triangle E^n(t, \sigma)$ is the instantaneous energy barrier along one sliding period. For sliding at constant velocity, $v$, we may replace the time coordinate with the spatial coordinate using $t = x/v$, yielding:

$$\frac{\mathrm{d}}{\mathrm{d}x}p(x, \sigma) = -\frac{f_0}{v} e^{\frac{-\triangle E^n(x,\sigma)}{k_B T}}p(x, \sigma). \quad (7)$$

Assuming a linear variation of $\triangle E^n(x, \sigma)$ between $\triangle E_{\max}^n(\sigma)$ and $\triangle E_{\min}^n(\sigma)$:

$$\triangle E^n(x, \sigma) = \triangle E_{\max}^n(\sigma) - \frac{x}{\triangle x}\left[\triangle E_{\max}^n(\sigma) - \triangle E_{\min}^n(\sigma)\right], \quad (8)$$

Equation (6) can be solved analytically, yielding (for an initial condition of $p(0, \sigma) = 1$):

$$p(x, \sigma) = e^{-c_1(\sigma)\left[e^{\frac{-\triangle E^n(x,\sigma)}{k_B T}} - e^{\frac{-\triangle E_{\max}^n(\sigma)}{k_B T}}\right]}, \quad (9)$$

where $c_1(\sigma) = f_0 k_B T\triangle x/\left\{\left[\triangle E_{\max}^n(\sigma) - \triangle E_{\min}^n(\sigma)\right]v\right\}$.

The probability density distribution of the dislocation to buckle at position $x$ along the sliding path is given by the corresponding reduction of the survival probability at this point, namely $f(x, \sigma) = -\mathrm{d}p(x, \sigma)/\mathrm{d}x$. With this, the energy dissipated by an individual dislocation due to shear induced buckling over a full sliding period is given by

$$\triangle w(\sigma) = \int_0^{\triangle x}\mathrm{d}x\left[\triangle E_{\max}^n(\sigma) - \triangle E^n(x, \sigma)\right]f(x, \sigma)H(\triangle E^n(x, \sigma)), \quad (10)$$

where $\triangle E_{\max}^n(\sigma) - \triangle E^n(x, \sigma)$ is the dissipated elastic energy invested in depressing the dislocation if it buckles at point $x$ and the Heaviside step function screens unphysical negative barrier heights. Note that this derivation follows the spirit of the Prandtl-Tomlinson model at finite temperatures[30–32]. The resulting expression for $\triangle w(\sigma)$ (see Supplementary Note 6) replaces the term $\triangle E_{\max}^n(\sigma)$ in Eq. (5) for the zero-temperature case and the calculation for the friction force proceeds accordingly. The results, appearing in Fig. 2c, d, show good correspondence with the simulation data, both for the normal load dependence at fixed temperature (Fig. 2c) and for the temperature dependence at fixed normal load (Fig. 2d). It should be noted that when fitting the model to the simulation results, the periodicity of the ILP profile along the sliding direction, $\triangle x$, was chosen to be 10.8 Å, ~1/3 of the periodicity $|\mathbf{T}| = \sim 3\,\mathrm{nm}$, reflecting the fact that dislocations may buckle more than once per sliding period (see Fig. 3c). Therefore, we conclude that the nonmonotonic frictional dependence on temperature originates from similar competing effects: (i) increase of buckling probability with temperature; and (ii) decrease in dissipated energy per buckling event when buckling occurs earlier along the path. The latter results from the fact that thermal fluctuations assist overcoming the barrier hence less energy needs to be invested (and lost) by the shear process in order to buckle.

## Discussion

Notably, one of the important assumptions underlying the phenomenological model is the independence of buckling events occurring at different dislocation centers. The good agreement that the model achieves with elaborate simulation results of various corrugated GBs indicates the validity of this assumption for realistic polycrystalline surfaces and undermines the importance of collective effects. This suggests that the frictional dissipation of corrugated GBs should be proportional to their overall length that, in turn, grows linearly with the overall contact area, $S_{tot}$. To demonstrate this, we note that the perimeter of an individual grain, i.e. the length of its GB, $L_{grain}$, is proportional to the square root of its surface area. Marking $P(S)$ as the probability density function of obtaining a grain of area $S$, the average grain area is given by $\bar{S}_{grain} = \int P(S)SdS$ and the corresponding average boundary length per grain is $\bar{L}_{grain} \propto \int P(S)\sqrt{S}dS$. The average number of grains is therefore given by $N_{grain} = S_{tot}/\bar{S}_{grain}$, yielding an average overall GB length of $L_{tot} \propto \zeta N_{grain}\bar{L}_{grain} = \zeta(\bar{L}_{grain}/\bar{S}_{grain})S_{tot}$, which is proportional to $S_{tot}$ for any reasonable distribution of grain areas. Here, $\zeta$ is a constant pre-factor of $O(1)$ accounting for overlapping boundaries of adjacent grains. Given this length scaling, the friction coefficient under a given normal pressure, $\sigma$, will be proportional to the GB length per unit area: $\mu \propto L_{tot}/S_{tot} \propto \zeta(\bar{L}_{grain}/\bar{S}_{grain})$. This stems from the fact that the friction coefficient is defined as the ratio between the friction and normal forces, $\mu = F_f/F_N$, where $F_f \propto L_{tot}$ and $F_N = S_{tot}\sigma$.

Aiming to scale-up structural superlubricity in layered material interfaces implies the inevitable appearance of GBs at the sliding interface. As shown above, the scaling of their dissipative contribution with contact area is stronger than the sublinear scaling found for pristine incommensurate layered contacts[29,33]. Nevertheless, the predicted nonmonotonic frictional dependence on normal load and temperature leads to negative differential friction coefficients at the high normal load regime. This, in turn, opens the way to achieve large-scale superlubricity by reducing the excess friction associated with each individual GB. The underlying mechanism, which is valid for any corrugated polycrystalline layered interface, involves two competing effects: an increase of buckling probability accompanied by a decrease of the dissipated energy per buckling event with normal load and/or temperature. This mechanism is quite different from simple steric hindrance considerations induced by rigid corrugated obstacles[34]. For incommensurate contacts, the contribution of flat GB geometries (where buckling events are absent) to the overall friction is small, as their tribological properties resemble those of a pristine interface. Therefore, the phenomenological model developed herein can serve as an efficient tool to characterize the frictional properties of complex interfaces consisting of involved GB geometries, as long as friction is dominated by buckling events. This may help reducing the dependence on explicit demanding atomistic simulations of large-scale polycrystalline mosaic interfaces and focusing computational efforts on understanding specific microscopic mechanisms of interest.

## Methods

The structure of the PolyGr layer is generated using a Voronoi tessellation method developed by Shekhawat and colleagues[35,36], which provides energetically favorable GBs with dislocation content and structural properties in excellent agreement with experimentally measured GBs[35], and introduces minor strain effect (see Supplementary Methods for details). Periodic boundary conditions are applied for all six layers along both lateral directions, thus mimicking an infinite interface (neglecting only flexural modes longer than the supercell considered). The intralayer and interlayer interactions are modeled with the second-generation reactive empirical bond order (REBO) potential[37] and the registry-dependent ILP[38–42] with the refined parameters[42] that provide accurate results up to high external pressures[43], respectively. This combination was extensively investigated in recent years and found to

provide very good agreement with experimental results of the compressibility[43], thermal conductivity[44], tribological properties[10], and phonon dispersion[43] of layered materials interfaces. Furthermore, this approach yields GB corrugations, topographies, and energies comparable to available experimental results and DFT calculations[15–18]. All simulations were performed using the LAMMPS package[45].

To remove the heat accumulated during sliding, velocity damping with a damping coefficient of $1.0\ ps^{-1}$ was applied to the relative velocities (with respect to the velocity of the rigid top slider layer) of each atom in the second layer from top (slider) and to the velocities of each atom in the fifth layer from top (substrate). For finite temperature simulations, Langevin thermostats (rather than mere viscous damping) are employed to these layers. This mimics the energy dissipation channels (via coupling to implicit external heat baths) through both slider and substrate in experiments, with minor effect on the dynamics of the two layers at the shear plane (see Supplementary Note 5). The thermostat is tested to be reliable in a wide range of sliding velocities and normal loads (see Supplementary Methods). In addition, in order to dampen vertical stage oscillations, velocity damping with the same damping rate of $1.0\ ps^{-1}$ is also applied to the vertical velocities of the atoms in the top slider layer[46]. See Supplementary Methods for further details regarding the simulation model and protocol.

## Data availability

The data that support the findings of the present study are available within the paper and its Supplementary file. Other data are available from the corresponding authors upon request.

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

## Acknowledgements

X.G. acknowledges the fellowship from the Sackler Center for Computational Molecular and Materials Science at Tel Aviv University. W.O. acknowledges the financial support from the starting-up fund of Wuhan University and the National Natural Science Foundation of China (No. 11890673 and No. 11890674). O.H. is grateful for the generous financial support of the Israel Science Foundation under Grant No. 1586/17 and the Naomi Foundation for generous financial support via the 2017 Kadar Award. M.U. acknowledges the financial support of the Israel Science Foundation, Grant No. 1141/18 and the binational program of the National Science Foundation of China and Israel Science Foundation, Grant No. 3191/19. M.U. and O.H. thank partial computational support of the Tel Aviv University Center for Nanoscience and Nanotechnology.

## Author contributions

O.H and M.U. designed research, X.G. conducted the research, X.G., W.O., M.U., and O.H. analyzed the data, and wrote the paper.

## Competing interests

The authors declare no competing interests.
