## [Peer Review File · Nature Communications]

Superlubric Polycrystalline Graphene InterfacesREVIEWER COMMENTS

Reviewer #1 (Remarks to the Author):

I have read the manuscript with interest and, although I found it to be well-written and attempting to address a very interesting question, I believe it to be unsuitable for publication in Nature Communications. The main reasons for this are:

(1) The paper claims the findings of this investigation to be general and to be applicable to "any large-scale polycrystalline layered interface". Although it is possible that the mechanism linked to the nonmonotonic frictional response of these polycrystalline layered systems could be linked to the snap-through buckling of dislocations observed in the atomistic model presented by the authors, in my opinion the authors have failed to prove that such effects are general and system independent. This is because there are various assumptions made in the simulations: starting from the system design, it is unclear why the 6 layers system used by the authors would be representative of a realistic scenario. The response of this system subjected to normal load and sliding is strongly affected by the stiffness of the system (hence number of layers and relative rigidity), the way the system is loaded, barostatted and thermostatted, by the applied sliding velocity and many other parameters, including the description of the grain boundaries. Physically-accurate and efficient barostatting and thermostating are key to these simulations as there is strong coupling between normal and shear response. The parameters and the models used for this are not tested in a broad range, which leaves the impression that their choice may be at least in part responsible for the observed response. Why is the hypothesis not tested for a system with e.g. a larger number of layers? Also, why was this only studied for a velocity of 5 m/s? Energy produced by frictional shear heating will obviously affect the response of the system and the simulations should have been conducted at different speeds not only to check the effect this has on the response (and on the phenomenological model) but also to check how adequate the thermostat is.

Again, there seems to be an interplay between increase of dislocation buckling probability and decrease of the dissipated energy per buckling event; it also makes sense that the energy associated to buckling is reduced when larger normal loads are applied but I remain to be convinced that this is not an artifact of the system set up and, more importantly, that the results are general and can be extended to all systems as claimed by the authors.

The phenomenological two-state model presented by the authors is underpinned by the evidence provided by the simulations, and, as such, is affected by the same potential limitations affecting the results from the atomistic simulations and their interpretation. There is an implication that the phenomenological model can be used to speed up design - how is this the case? This study only offers a way to speed up analysis, not a solution to a problem in case of e.g. application of low loads to these layered systems.

(2) There is no experimental evidence corroborating the findings from the modelling exercise conducted by the authors; the claim of generality of results contrasts sharply with the fact that the authors do not even attempt to relate any of their results to experimental investigations. How would their system compare with a realistic experiment? Without a thorough study that includes expansion of the models to larger systems and a larger parameter space (see above) the alleged mechanisms responsible for the nonmonotonic friction response of these systems with load remains an unproven hypothesis.

Reviewer #2 (Remarks to the Author):

This article studied the friction of polycrystalline graphene layers and found that the friction force exhibited an unusual dependence on the normal force and temperature, especially they observed the negative differential friction coefficients. The authors conjectured that this behavior is caused by a dynamic "snap-through buckling" between an upward protrusion and a downward protrusion of GB dislocations. They constructed a phenomenological model based on the first-order rate equation with two states. These findings are very interesting and of critical importance in the study of superlubricity on the macroscopic scales and the agreement between the model predictions and the simulation results look excellent. One concern is how we know whether the material behaviors observed from their MD simulations are real. Certainly, the material properties and behaviors in an

atomistic simulation are governed by the interatomic potential used in the simulation and an interatomic potential is just an approximation of the material behavior under certain and limited circumstances. Therefore, I think the authors need to augment the article by adding some experimental evidence or a verification from more reliable models, such as DFT, supporting their simulation observations.

Reviewer #3 (Remarks to the Author):

Review of Gao, Ouyang, Urbakh, Hod

The authors present a combination of molecular dynamics simulations and phenomenological modeling to explore and explain nonmonotonic dependences of friction on both temperature and compressive normal stress for graphene sliding against graphene, where one of the graphene layers has defects in the form of two grain boundaries. If there were no defects, extremely low friction (“structural superlubricity”) would be expected since the graphene layers include an incommensurate contact. However, the defects can inhibit this effect. This is important to study scientifically since there is little understanding or fundamental theory to predict or explain how defects affect frictional sliding.

The authors observe that the presence of the grain boundaries leads to a rise and then decrease in friction vs. temperature, and vs. stress. They explain this as resulting from the presence of two competing effects: shear-induced local out-of-plane buckling of clusters atoms (typically 5-7 Stone-Wales defects) along the grain boundaries (leading to an increase in friction with temperature at low temperatures since thermal energy increases the number of such buckled dislocation), and then thermally-induced frequent buckling (leading to a subsequent decrease in friction with temperature). Compressive stress reduces the barrier between the two buckled states, leading to a nonmonotonic dependence on stress as well.

The problem studied has practical interest, since the shearing of graphite layers across each other has been shown in experimental studies to exhibit extremely low friction (“superlubricity”), but defects like grain boundaries can limit the scale-up of this effect to larger sizes, where such defects can increase friction significantly. The authors argue these results can aid in avoiding this problem.

The methodology used is sound, and it is impressive that the relatively simple phenomenological method reproduces multiple trends found in the simulations very well. The extraction of the energy barrier profile from the fluctuation statistics (Fig 3a) is also particularly nice. As well it is innovative that the work proposes a new, previously unappreciated mechanism for frictional dissipation of layered materials. This is an advance over many previous efforts which typically consider perfect, defect-free systems. Considering the complex role of defects is important in realizing scaled-up, practical realizations of low friction interfaces.

In general the manuscript is well-written. However, some points were not clearly presented, and it is recommended that the authors revise the manuscript to address the following points:

Major Points

1. The authors discuss “shear induced buckling” and “shear induced out-of-plane motion” at the grain boundaries, but they do not ever explain how shear causes buckling. They show clear illustrations of the buckling itself, but it is not clear how a surface shearing over such a defect leads to buckling. Is it because the upper defect-free surface pushes on an upward-pointing buckled structure in the downward direction? If so, then why does it recover? The simple kinematic and mechanical origin of this effect, which is one of the fundamental unit processes considered in this article, is simply never explained or clearly defined. This could be done on page 5, in the paragraph starting at line 135. The authors should also explain what they mean by “variation of the snap-through barrier along the scan line”. The referee does not understand what they mean; does this barrier vary with position somehow?

2. The abstract is very unclear and vague in certain ways. It should specify that the system studied is for flat graphene sliding on flat graphene layers (i.e. not a tip on graphene for example); that MD is used to obtain the results; that they are studying kinetic, not static friction; and that they are simulating at 5 m/s (to distinguish the work from, e.g., AFM-based studies).
3. The authors do not actually show that their interfaces are superlubric. The definition of superlubricity is either a very low absolute (not differential) friction coefficient (<0.01), or a very low shear strength. Here, the friction forces are in the range of 0.01-0.05 nN. Using the smaller end of this range, I calculate a shear stress of approximately 30 kPa, which is indeed impressively low (assuming the area is as given in Supp. Fig. 1). Even at 300K and zero stress, friction force is approx. 0.04, or 120 kPa. This should be specified in the manuscript and compared to experimental measurements, which are available.
4. The size of the system is not given in the main text, making it hard to connect forces and stresses. Adding the dimensions to Fig. 1 (as are shown in Supp. Fig. 1) would be helpful.

Minor Points:

5. Also, the phrase in the abstract "where an increase of dislocation buckling probability is accompanied by a decrease of the dissipated energy per buckling event" is a bit confusing since "accompanied by" suggests the two effects occur together, when in fact one dominates the other. Perhaps "competes with" would work better
6. Fig 1d would be clearer if the authors labeled the different layers in with their angles (i.e. add a label showing that the top 3 are at 38.2° ; the next one is at 0° and 8° , the bottom at 0°). As well, adding the layer number to Fig. 1a and 1d would be helpful. Related to this, the author found it hard to visualize the description in the first two paragraphs under "Results" since the shear plane not specified in Fig. 1. A label or line clearly showing the shear plane should be added to Fig 1a and 1d. Also, the "slider" and the "substrate" should be clearly labeled in Fig. 1a and 1d.
7. A movie showing all 5 layers, illustrated in a 3d perspective view, would be extremely helpful. This way, one could see the sliding interface. The movie could start by showing all 5 layers, then repeated with the top 3 layers removed so one can see the buckling.
8. The authors may wish to note that the pentagon-heptagon defects are known also as a type of Stone-Wales defects in the literature.

In what follows we provide a detailed point-by-point response to all concerns raised by the three reviewers.

Reviewer #1

"I have read the manuscript with interest and, although I found it to be well-written and attempting to address a very interesting question, I believe it to be unsuitable for publication in Nature Communications"

We thank the reviewer for his/her thorough consideration of our manuscript. We further thank him/her for stating that our work is well-written and addresses an interesting question. We humbly disagree with the reviewer's evaluation that our paper is unsuitable for *Nature Communications*. In what follows we provide detailed explanations and additional simulation results supporting our claim by proving the generality and system independence of our predictions.

"The paper claims the findings of this investigation to be general and to be applicable to "any large-scale polycrystalline layered interface". Although it is possible that the mechanism linked to the nonmonotonic frictional response of these polycrystalline layered systems could be linked to the snap-through buckling of dislocations observed in the atomistic model presented by the authors, in my opinion the authors have failed to prove that such effects are general and system independent. This is because there are various assumptions made in the simulations: starting from the system design, it is unclear why the 6 layers system used by the authors would be representative of a realistic scenario. The response of this system subjected to normal load and sliding is strongly affected by the stiffness of the system (hence number of layer and relative rigidity)" and "Why is the hypothesis not tested for a system with e.g. a larger number of layers?"

The reviewer raises an important concern regarding the adequacy of our model system in terms of its thickness. To address these issues, we performed additional simulations considering thicker model systems consisting of 8 and 10 layers (only one of which is PolyGr), half residing in the slider and half in the substrate. The geometry of each model system was first optimized and then annealed as done for the 6-layered stack. During the dynamics, damping was applied to the two layers adjacent to the outer fixed substrate layer and rigid slider layer (the second layer from the top and the second layer from the bottom of the stack). To reduce computational burden, the comparison was performed only at zero temperature.

Fig. R1(a) below presents the annealed corrugated topographies of the PolyGr layer for different model system thicknesses, which are found to be very similar in all three cases considered. Minor differences in the protrusion direction may be found depending on the specific annealing realization. Fig. R1(b) demonstrates that the load dependence of the friction is weakly sensitive to the model system thickness. Furthermore, similar dislocation buckling trajectories are obtained for different model system thicknesses (see Fig. R1(c)-(d)). This demonstrates that the 6-layered model system used to obtain the results presented in the main text can be utilized to study the frictional behavior of thicker stacks incorporating a PolyGr layer.

To address this issue, we have added a discussion in Supplementary Note 1 (see page 4 of the revised SI), we have incorporated the new Supplementary Note 6, and added a corresponding discussion on page 7 of the revised main text.

Fig. R1. Comparison of simulation results for model systems of different thicknesses. (a) Annealed topographies of the PolyGr layer embedded within stacks of various number of layers. The color represents the atomic out-of-plane deformation (see color scale). (b) Load dependence of the friction

force at zero temperature calculated for the 6- (black squares), 8- (red circles), and 10-layered (blue triangles) systems. (c)-(d) Representative dislocation trajectories taken from the (c) 8-layered system and the (d) 10-layered system, respectively, under normal loads of 0, 0.6, and 1.9 GPa (from top to bottom). The dislocations in each panel (different line colors) are chosen to be the same as those appearing in Fig. 3(c) of the main text.

“the way the system is loaded, barostatted and thermostatted, by the applied sliding velocity and many other parameters, including the description of the grain boundaries. Physically-accurate and efficient barostatting and thermostating are key to these simulations as there is strong coupling between normal and shear response. The parameters and the models used for this are not tested in a broad range, which leaves the impression that their choice may be at least in part responsible for the observed response.”

The reviewer raises an important concern regarding the adequacy of our model system in terms of the loading, thermostating and barostatting protocols.

Regarding the way that the external load is modeled in our simulations: The normal load is applied by adding a uniform force in the range of 0-0.06 nN in the vertical direction, to each atom in the topmost rigid layer corresponding to normal pressures of up to ~ 2.3 GPa. This procedure mimics a realistic scenario where a uniform load spreads from the bulk region of a sufficiently thick slider to the shear interface, where the load distribution can be nonuniform due to surface corrugations. We note that the lateral dimension change of the graphene layers under normal load can be neglected due their extremely high in-plane stiffness. This approach is commonly used in MD simulations of sheared interfaces (Ye et al., Phys. Rev. B 2017, 96, 115401; Ye et al., Phys. Rev. B 2016, 93, 235438; Mandelli et al., Sci. Rep. 2017, 7, 10851; Gao et al, Phys. Rev. B 2021, 103, 045418). Furthermore, the fact that we obtain practically the same frictional behavior for three loaded model system thicknesses indicates that the buffer layers residing between the loaded upper surface and the sheared interface is sufficiently thick to provide converged results.

Regarding the reviewer’s comment on barostatting: we deliberately refrained from employing a Nosé–Hoover type barostat, which couples to the dimensions of the simulation box, during our simulations for the following reasons: (i) In the vertical (out-of-plane) direction, our simulation box includes a large vacuum region to avoid spurious interactions between image stacks. The Nosé–Hoover barostat is not well suited to regulate the pressure in such a configuration mimicking an open system. Instead, we regulate the pressure by applying the external normal force to the top stack layer; (ii) In the lateral directions the system size is chosen such that the introduced strain is below 1‰. Due to the extremely high in-plane stiffness of graphene, variations of the box lateral dimensions under load can be readily neglected.

Considering the adequacy of the thermostating procedure: we employed Langevin thermostates to the layers adjacent to the topmost rigid and lowest fixed layers within the stack. The damping coefficient

was chosen as 1.0 ps^{-1} , which was previously shown to have minor effect on the calculated friction in layered materials interfaces (Gao et al., Phys. Rev. B 2021, 103, 045418). The target temperature is achieved by introducing random forces, which satisfy the fluctuation-dissipation theorem. These random forces, applied to different atoms, are independent, thus the total random force on the each thermostated layer is not exactly zero at each time step. While over infinite simulation time this will be averaged out, at finite time this may introduce a non-zero center-of-mass force on the thermostated layers. Since the overall friction is very small, such a non-physical temporary force may jeopardize the physical relevance of the simulation results. Therefore, we apply the standard procedure implemented in LAMMPS, where at each time step the vectorial sum of all atomic random forces within a given thermostated layer (divided by the number of atoms) is subtracted from the force acting on each atom in this layer. As shown in Fig. R2, this thermostat scheme maintains the target temperatures well under different normal loads and sliding velocities.

Fig. R2. Representative instantaneous temperature profiles in the two thermostated layers under different normal loads, target temperatures, and sliding velocities. Temperature profiles for (a)-(c) utilize a target temperature of 50 K under different normal loads [(a) 0 GPa, (b) 0.4 GPa, and (c) 0.8 GPa] with a sliding velocity of $v_0=5 \text{ m/s}$; (d)-(f) different target temperatures [(d) 150 K, (e) 300K, and (f) 400K] under a normal load of 0.4 GPa and a sliding velocity of 5 m/s sliding velocity, (g)-(i) different sliding velocities ((g), (h) 2 m/s and (i) 10 m/s) and a target temperature of (g), (i) 50K and (h) 150K under zero normal

load. T_2 and T_5 represent the temperatures in the second layer and the fifth layer from top, where the Langevin thermostates are applied. The blue dashed lines denoting the target temperatures are shown as a guide to the eye.

Regarding the description of grain boundaries (GBs), as explained in the text, we employ a Voronoi tessellation method developed by Shekhawat (Phys. Rev. B 2015, 92, 205402) to generate the GB structure. Compared to annealing two inclined graphene lattices to form a GB, this method gives considerably lower GB energy. More importantly, the GBs generated by this method show excellent agreement with experimental observations in dislocation content and GB structures. For the description of intra- and inter-layer interactions, the second-generation reactive empirical bond order (REBO) potential (Brenner et al., J. Phys.: Condens. Matter 2002, 14, 783–802), which yields physical properties for graphene in satisfactory agreement with DFT and experimental references (Lebedeva et al. Physica E Low Dimens. Syst. Nanostruct. 2019, 108, 326–338), and the dedicated registry-dependent interlayer potential, developed by our and other groups, parameterized against state-of-the-art DFT calculations (Kolmogorov and Crespi, Phys. Rev. B 2005, 71, 235415; Ouyang et al, Nano Lett. 2018, 18, 6009–6016; Ouyang et al., J. Chem. Theory Comput. 2020, 16, 666–676), showing also good agreement in physical properties with experiments, e.g. compressibility, bulk moduli, binding energy, and all relevant phononic effects, up to very high pressures (~30 GPa). The combination of these potentials gives GB corrugations, topography and GB energies comparable to experiments and DFT calculations (Gao et al, Phys. Rev. B 2021, 103, 045418; Červenka & Flipse Phys. Rev. B 2009, 79, 195429; Tison et al. Nano Lett. 2014, 14, 6382–6386; and Yazyev and Louie Phys. Rev. B 2010, 81, 195420).

To clarify these points, we added a short discussion in the Methods section of the revised manuscript supporting the relevance of our model systems to experimentally studied GBs, and in Supplementary Note 1, under the item titled “simulation protocol”, we added a discussion on the adequacy of the loading protocol and the inadequacy of using a Nosé–Hoover barostat in our simulation setup. We further added in this supplementary section a numerical demonstration of the validity of our thermostating procedure under different normal loads and sliding velocities, with appropriate reference in the Method section.

“Also, why was this only studied for a velocity of 5 m/s? energy produced by frictional shear heating will obviously affect the response of the system and the simulations should have been conducted a different speed not only to check the effect this has on the response (and on the phenomenological model) but also to check how adequate the thermostat is.”

The reviewer raises an important concern regarding the adequacy of our model system in terms of the studied velocity. To address this important point, we have repeated our simulations at two additional

velocities of 2 m/s and 10 m/s. Similar to the case of a sliding velocity of 5 m/s, dynamic snap-through dislocation buckling is found (see Fig. R3), leading to a nonmonotonic to monotonic transition of the friction load dependence with increasing temperature (see Fig. R4). Increasing the sliding velocity results in a mild shift of the friction force peak position toward higher normal loads. This results from the fact that at higher velocities the timescale for each individual buckling event is reduced, thus reducing the probability of thermal activation to promote buckling. In such case, a higher normal load is required for buckling to occur. Such behavior is also reproduced by our phenomenological model. The fact that we find only mild quantitative (rather than qualitative) changes of the system's friction with sliding velocity suggests that the mechanism underlying the frictional behavior predicted by our simulations is insensitive to variations of the sliding velocity within the wide velocity range considered.

To clarify this point we added a discussion on page 7 of the revised main text referring the reader to the velocity dependence study presented in Supplementary Note 6 of the revised manuscript.

Fig. R3. Representative GB dislocation vertical trajectories for sliding velocities of (a) $v_0 = 2$ m/s and (b) $v_0 = 10$ m/s at zero temperature and under normal loads of 0, 0.6, and 1.9 GPa (from top to bottom). The trajectories correspond to the same GB dislocations as in Fig. 3c of the main text.

Fig. R4. Load dependence of the friction force obtained for sliding velocities of 2, 5, and 10 m/s. (a)-(c) MD simulation results. (d)-(f) Model predictions. The same parameter set as in Fig. 2(c)-(d) of the main text is used.

“Again, there seems to be an interplay between increase of dislocation buckling probability and decrease of the dissipated energy per buckling event; it also makes sense that the energy associated to buckling is reduced when larger normal loads are applied but I remain to be convinced that this is not an artifact of the system set up and, more importantly, that the results are general and can be extended to all systems as claimed by the authors.”

The reviewer raises two important concerns one regarding the adequacy of our model systems and the other regarding the general nature of our main conclusions. The former was already addressed in detail above, where we proved that our predictions are not an artifact of the system setup and that the unveiled frictional mechanism appears for thicker and stiffer model systems as well as for different slider velocities.

To demonstrate that the GB dislocation buckling frictional mechanism is not limited to the specific model system discussed in the main text, we have repeated our simulations for two other systems:

1. The first system is a homogeneous polycrystalline graphitic interface, similar to the one discussed in the main text, but with a smaller GB misfit angle of $\theta_2 = 2.5^\circ$ that exhibits higher GB corrugation. For this system, we find that the overall qualitative dependence of the friction

on the normal load and temperature is preserved with some quantitative modifications due to the increased barrier heights (see Fig. R5 and Supplementary Note 6 in the revised manuscript).

2. In the second system we have extended our treatment to heterogeneous junctions, considering an interface between PolyGr and *h*-BN layers. Dynamic snap-through buckling of GB dislocations is also found for this system leading to similar nonmonotonic load dependence of the friction force (see Fig. R6). These results indicate that the frictional behavior observed and the underlying snap-through buckling dynamics are general to polycrystalline layered interfaces with corrugated GBs, across homo- and hetero-junctions. Since these results are part of a future research and are beyond the scope discussed herein, focusing on homogeneous polycrystalline interfaces, we opted not to present them in the current paper.

Fig. R5. Model system and load-dependent friction force for a PolyGr junction with a misfit angle of $\theta_2 = 2.5^\circ$. (a) Top view topographic map of the annealed PolyGr layer within the six layered stack (the top three layers are not shown). False atom coloring represents the relative atomic height with respect to the average height of the grains (see color bar). (b) Load dependence of the friction force of this GB at various temperatures.

Fig. R2. Load dependence of the friction force in an *h*-BN/PolyGr heterojunction. (a) Load dependence of friction at zero temperature and at 300 K. (b) Representative vertical GB dislocation trajectories under different normal loads.

To clarify this point we added a discussion on page 7 of the revised main text referring the reader to the analysis of the frictional properties of the higher corrugated GB presented in Supplementary Note 6 of the revised manuscript.

“The phenomenological two-state model presented by the authors is underpinned by the evidence provided by the simulations, and, as such, is affected by the same potential limitations affecting the results from the atomistic simulations and their interpretation. There is an implication that the phenomenological model can be used to speed up design - how is this the case? This study only offer a way to speed up analysis, not a solution to a problem in case of e.g. application of low loads to these layered systems.”

We thank the reviewer for raising this point. First, we would like to note that in our response to the above comments made by the reviewer, we have addressed in detail her/his concerns regarding the limitations of our modeling approach and the general nature of our predictions. Therefore, while the phenomenological two-state model is indeed constructed based on the evidence of the numerical simulations, it does not suffer from the implied lack of generality.

Regarding the reviewer’s questions about the role of the phenomenological model in designing large-scale polycrystalline interfaces with desired frictional properties, we stress that the analytical treatment allows us to extend the study towards systems that are beyond current molecular dynamics simulation capabilities. Specifically, it allows us to study large-scale polycrystalline layered materials interfaces

with complex GB morphologies, by studying the local environment of individual GB dislocations using fully atomistic simulations and using the results as an input for the phenomenological model (similar to what was done in the present study). Furthermore, the model enables us to investigate parameter regimes that are difficult to simulate, such as low sliding velocities (relevant to many experimental conditions) and a broad range of loads. With respect to the latter, we stress that the load dependence enters the phenomenological model explicitly via its influence on the barrier height.

“There is no experimental evidence corroborating the findings from the modelling exercise conducted by the authors; the claim of generality of results contrasts sharply with the fact that the authors do not even attempt to relate any of their results to experimental investigations. How would their system compare with a realist experiment?”

This point made by the referee is quite controversial and goes against the accepted scientific paradigms completely undermining the central role of computational and theoretical science. Our manuscript presents a computational and theoretical prediction (rather than a “modeling exercise”) that can and should be confronted with future experimental efforts. However, the fact that it is presented as a stand-alone prediction cannot serve to argue against its suitability for publication. In fact, under the measure that every theoretical prediction should appear simultaneously with existing experiments, many breakthrough scientific discoveries in general, and in the field of tribology in particular, would not have been published. To name a few, the Aubry transition (Aubry, *Physica D* 1983, 7, 240–258), the Frenkel-Kontorova model (Kontorova, Frenkel, *Phys. Z. Sowietunion* 1938, 13, 1.), and even our own prediction of robust superlubricity in layered materials heterojunctions - a theoretical prediction (Leven, et al., *J. Phys. Chem. Lett.* 2013, 4, 115.) that was later validated and expanded on by several experimental groups (Song, et al., *Nature Mater.* 2018, 17, 894–899; Liu et al., *ACS Nano* 2018, 12, 7638–7646). We therefore strongly believe that our manuscript is suitable for publication in *Nature Communications*, a venue of broad impact and wide readership that can stimulate new and exciting experiments and applications based on our computational and theoretical predictions.

Our simulation model system includes a sliding interface that consists of a polycrystalline graphene layer. This layer is supported by a sufficiently thick substrate model and sheared against a sufficiently thick slider model both providing converged results (see our responses above). The inter-atomic interactions within the layer are modeled by well accepted and heavily studied classical force fields, that provide a good qualitative descriptions of the mechanical properties of graphene and *h*-BN. The interlayer interactions are described by dedicated interlayer potentials, that are currently the state-of-the-art in treating layered materials interfaces, providing both qualitative and quantitative agreement with advanced ab-initio reference data. This combination of model systems and force fields, have been extensively investigated in recent years and found to provide very good agreement with experimental results of the compressibility (Ouyang et al., *J. Chem. Theory Comput.* 2020, 16, 666–676), thermal

conductivity (Ouyang et al., *Nano Lett.* 2020, 20, 7513–7518), tribological properties (Song, et al., *Nature Mater.* 2018, 17, 894–899), and phonon dispersion (Ouyang et al., *J. Chem. Theory Comput.* 2020, 16, 666–676) of layered materials interfaces.

To clarify this point we added a short discussion in the Method section providing references to the abovementioned studies that support the validity and suitability of our simulation approach.

“Without a thorough study that includes expansion of the models to larger systems and a larger parameter space (see above) the alleged mechanisms responsible for the nonmonotonic friction response of these systems with load remains an unproven hypothesis.”

Reiterating the above detailed discussion, we have expanded our parameter space to include a wide spectrum of velocities and thicker model systems. We have further extended our study, considering different types of GBs as well as heterogeneous layered materials interfaces. In terms of system dimensions, we note that due to the cyclic boundary conditions used, our model system actually mimics an infinite interface, neglecting only flexural modes longer than the unit-cell considered.

To clarify this point, we added in the Methods section a comment stating that: “Periodic boundary conditions are applied for all six layers along both lateral directions, thus mimicking an infinite interface (neglecting only flexural modes longer than the unit-cell considered).”.

Reviewer #2

"This article studied the friction of polycrystalline graphene layers and found that the friction force exhibited an unusual dependence on the normal force and temperature, especially they observed the negative differential friction coefficients. The authors conjectured that this behavior is caused by a dynamic "snap-through buckling" between an upward protrusion and a downward protrusion of GB dislocations. They constructed a phenomenological model based on the first-order rate equation with two states. These findings are very interesting and of critical importance in the study of superlubricity on the macroscopic scales and the agreement between the model predictions and the simulation results look excellent."

We thank the reviewer for her/his positive evaluation of our manuscript.

"One concern is how we know whether the material behaviors observed from their MD simulations are real. Certainly, the material properties and behaviors in an atomistic simulation are governed by the interatomic potential used in the simulation and an interatomic potential is just an approximation of the material behavior under certain and limited circumstances. Therefore, I think the authors need to augment the article by adding some experimental evidence or a verification from more reliable models, such as DFT, supporting their simulation observations."

We thank the reviewer for raising this important point, which gives us the opportunity to provide a clearer explanation of this issue. The reviewer raises concerns regarding the relevance of the predictions of our classical MD simulations to realistic experimental scenarios. With this respect, we agree with the reviewer's statement that: "Certainly, the material properties and behaviors in an atomistic simulation are governed by the interatomic potential used in the simulation and an interatomic potential is just an approximation of the material behavior under certain and limited circumstances." Below, we justify the choice of force-fields as a powerful tool to predict new phenomena.

Layered materials interfaces are characterized by highly anisotropic interactions including strong intralayer covalent bonding and weaker Van der Waals interlayer interactions. As such, standard isotropic force fields fail to simultaneously describe both binding and shearing properties of such junctions. To this end, dedicated anisotropic classical force-fields have been developed (Kolmogorov and Crespi, Phys. Rev. B 2005, 71, 235415) that provide the desired balance between computational accuracy and burden in the study of the tribological properties of layered materials interfaces. Our choice of combination of intralayer and interlayer force-fields, model systems, and simulation protocols, is therefore carefully made based on the adequate dedicated force field construction and our previous experience of its ability to rationalize experimental observation and predict new phenomena.

Specifically, for the description of the intralayer interactions within each graphene layer, we used the second-generation reactive empirical bond order (REBO) potential, which yields physical properties such as bond length, Young's modulus, bending rigidity, in satisfactory agreement with DFT and experimental references (Brenner et al., *J. Phys.: Condens. Matter* 2002, 14, 783–802; Lebedeva et al. *Physica E Low Dimens. Syst. Nanostruct.* 2019, 108, 326-338). For the description of the interlayer interactions, we employed the dedicated registry-dependent interlayer potential, developed by our groups and others and carefully calibrated against experimental data and state-of-the-art density functional theory calculations (e.g. Kolmogorov and Crespi, *Phys. Rev. B* 2005, 71, 235415; Ouyang et al, *Nano Lett.* 2018, 18, 6009–6016; Ouyang et al., *J. Chem. Theory Comput.* 2020, 16, 666–676). In particular, in our DFT reference calculations, we used the screened hybrid HSE06 exchange-correlation density functional approximation augmented by advanced many-body dispersion (MBD) corrections - a combination that provides very good agreement with binding energies and sliding energy corrugations obtained by higher accuracy DFT methods, such as the quantum Monte-Carlo approach and the random phase approximation, as well as available experimental data (Mostaani et al., *Phys. Rev. Lett.*, 2015, 115, 115501; Spanu et al., *Phys. Rev. Lett.* 2009, 103, 196401; Lebègue, et al., *Phys. Rev. Lett.* 2010, 105, 196401; Zacharia et al., *Phys. Rev. B*, 2004, 69, 155406; Wang et al., *Nat. Commun.* 2015, 6, 7853). The dedicated force fields were further shown to capture experimental results of the compressibility (up to very high pressures of ~30 GPa), bulk moduli, binding energies, and the low-energy phonon spectrum of graphitic interfaces. (Ouyang et al., *J. Chem. Theory Comput.* 2020, 16, 666–676) Furthermore, the force field successfully predicted unique frictional phenomena later validated by experiments (Mandelli et al., *Sci. Rep.* 2017, 7, 10851; Song et al., *Nature Mater* 2018, 17, 894–899). Importantly, in the context of the present study, the choice of intra- and inter-layer potentials and GB model systems (Gao et al, *Phys. Rev. B* 2021, 103, 045418), was shown to yield GB corrugation and topography that are comparable to available experiments observations (Červenka & Flipse, *Phys. Rev. B* 2009, 79, 195429; Tison et al. *Nano Lett.* 2014, 14, 6382-6386). In addition, the obtained GB energetics agrees well with that obtained via DFT calculations (Yazyev and Louie, *Phys. Rev. B* 2010, 81, 195420).

To further verify the relevance of our model system and choice of parameters, we have added to the revised manuscript a convergence test of our results with respect to the stack thickness and sliding velocity, demonstrating that the qualitative nature of predictions is independent of the choice of these parameters over a wide and physically relevant parameter range.

Based on all of the above, we are confident that the predictions made, based on our simulation results and the corresponding phenomenological model, are reliable and relevant to realistic experimental scenarios. As such, we believe that our study will stimulate new and exciting experiments and applications of polycrystalline layered materials superlubric interfaces.

To emphasize this point, we have added the following discussion in the Methods section of the revised main text: “The intralayer and interlayer interactions are modeled with the second-generation reactive empirical bond order (REBO) potential³⁸ and the registry-dependent interlayer potential³⁹⁻⁴³ with the refined parameters⁴³ that provide accurate results up to high external pressures⁴⁴, respectively. This

combination was extensively investigated in recent years and found to provide very good agreement with experimental results of the compressibility⁴⁴, thermal conductivity⁴⁵, tribological properties¹⁰, and phonon dispersion⁴⁴ of layered materials interfaces. Furthermore, this approach yields GBs corrugations, topographies, and energies comparable to available experimental results and DFT calculations¹⁵⁻¹⁸.”

Reviewer #3

“The authors present a combination of molecular dynamics simulations and phenomenological modeling to explore and explain nonmonotonic dependences of friction on both temperature and compressive normal stress for graphene sliding against graphene, where one of the graphene layers has defects in the form of two grain boundaries. If there were no defects, extremely low friction (“structural superlubricity”) would be expected since the graphene layers include an incommensurate contact. However, the defects can inhibit this effect. This is important to study scientifically since there is little understanding or fundamental theory to predict or explain how defects affect frictional sliding.

The authors observe that the presence of the grain boundaries leads to a rise and then decrease in friction vs. temperature, and vs. stress. They explain this as resulting from the presence of two competing effects: shear-induced local out-of-plane buckling of clusters atoms (typically 5-7 Stone-Wales defects) along the grain boundaries (leading to an increase in friction with temperature at low temperatures since thermal energy increases the number of such buckled dislocation), and then thermally-induced frequent buckling (leading to a subsequent decrease in friction with temperature). Compressive stress reduces the barrier between the two buckled states, leading to a nonmonotonic dependence on stress as well.

The problem studied has practical interest, since the shearing of graphite layers across each other has been shown in experimental studies to exhibit extremely low friction (“superlubricity”), but defects like grain boundaries can limit the scale-up of this effect to larger sizes, where such defects can increase friction significantly. The authors argue these results can aid in avoiding this problem.

The methodology used is sound, and it is impressive that the relatively simple phenomenological method reproduces multiple trends found in the simulations very well. The extraction of the energy barrier profile from the fluctuation statistics (Fig 3a) is also particularly nice. As well it is innovative that the work proposes a new, previously unappreciated mechanism for frictional dissipation of layered materials. This is an advance over many previous efforts which typically consider perfect, defect-free systems. Considering the complex role of defects is important in realizing scaled-up, practical realizations of low friction interfaces.

In general, the manuscript is well-written.”

We thank the reviewer for her/his detailed summary and positive evaluation of our work.

Major points of Reviewer #3:

“1. The authors discuss “shear induced buckling” and “shear induced out-of-plane motion” at the grain boundaries, but they do not ever explain how shear causes buckling. They show clear illustrations of the buckling itself, but it is not clear how a surface shearing over such a defect leads to buckling. Is it because the upper defect-free surface pushes on an upward-pointing buckled structure in the downward direction? If so, then why does it recover? The simple kinematic and mechanical origin of this effect, which is one of the fundamental unit processes considered in this article, is simply never explained or clearly defined. This could be done on page 5, in the paragraph starting at line 135. The authors should also explain what they mean by “variation of the snap-through barrier along the scan line”. The referee does not understand what they mean; does this barrier vary with position somehow?”

We thank the reviewer for pointing out this very important issue, which made us realize that we failed to provide a satisfactory explanation of the shear-induced buckling concept. As the referee states, one might envision that an “ironing” effect of the protrusion dominates the buckling process. Such an effect is indeed clearly seen when considering finite flakes sliding over individual grain boundary protrusions (Gao et al., Phys. Rev. B 2021, 103, 045418). In the case of infinite periodic interfaces, studied herein, the upper layer constantly pushes down on the upward protruding grain boundary defects due to the external normal load. The dominant effect, in this case, is the variation of the energy profile (specifically the barrier height) along the buckling trajectory with the lateral displacement of the slider (see Figs. R7 and R8). These variations stem from the fact that the interlayer Pauli repulsions between electronic clouds associate with atoms residing on adjacent layers (and hence the ILP) are registry dependent. At certain positions (e.g. the purple line in Fig. R7) the energy barrier along the buckling trajectory vanishes allowing for buckling to occur even at zero temperature. Further variations of the buckling energy profile during sliding may result in reverse buckling. This is manifested as dynamic dislocation buckling during sliding. The fact that phenomenological model that we devised based on this picture captures well the simulation results, indicates that indeed the variation of the buckling energy barrier with the relative displacement of the sheared layers is the dominant effect. We note that some contribution to the buckling probability may come from local heating at the vicinity of the buckling dislocations. However, since our phenomenological model nicely reproduces the atomistic simulation results without including local heating effects, indicates that under the simulation conditions, they have a minor contribution to the buckling process.

To clarify this point, we have made the following changes in the main text and SI:

On page 5 of the revised main text, we have added the following discussion: “Importantly, the energy profile along the vertical dislocation buckling trajectory varies with the lateral displacement of the slider (see Fig. S13 of Supplementary Note 5). These variations stem from the fact that, by construction, the adopted classical interlayer potential (ILP, see Methods section) accounts for the interlayer Pauli repulsions between electronic clouds associate with atoms residing on adjacent layers, which are registry dependent. Therefore, at certain positions (e.g. the purple line in Fig. S13b) the energy barrier along the buckling trajectory vanishes allowing for buckling to occur even at zero temperature. Further variations of the buckling energy profile during sliding may result in reverse buckling, thus manifesting dynamic dislocation buckling during sliding.”

On pages 6 of the revised main text, the following sentence was added: “The former results from the registry dependence of the vertical buckling energy profile, as discussed above, whereas the latter, which may be caused by local heating, is found to be of minor importance under our simulation conditions, as demonstrated below by comparison to a phenomenological two-state model.”

On page 23 of the SI, the title of Note 5 was modified to “Additional Information for Transition Energy Barriers of Dislocations”. The former title is now used for the first subsection in Note 5.

On page 25 of the SI, the subsection with title “Variation of the Buckling Energy Profile with the Lateral Displacement of the Slider” was added, which contains the following content: “Supplementary Fig. S13 shows the variation of the energy profile along the buckling trajectory with the lateral displacement of the slider for four representative dislocations. These variations stem from the fact that the interlayer interactions (and hence the ILP) are registry dependent (see several representative stacking modes in Supplementary Fig. 14). At certain positions (e.g. the purple line in Supplementary Fig. 13b) the energy barrier along the buckling trajectory vanishes allowing for buckling to occur even at zero temperature. Further variations of the buckling energy profile during sliding may result in reverse buckling. This is manifested as dynamic dislocation buckling during sliding.”

On page 25 of the SI, Figs. R7 and R8 have been added as Supplementary Figs. 13 and 14.

Fig. R7. Buckling free energy profiles of four representative dislocations (a)-(d) at six equidistant slider positions denoted by the numbers 1-6 along one sliding period of 29.8 \AA , under a normal load of 0.4 GPa . The buckling energy profile at position 1 in panel (a) is the same as that presented in Fig. 4(a) of the main text.

Fig. R8. Illustration of the equidistant slider positions used for the free energy profile calculations. (a)-(f) The stacking configurations for six equidistant slider positions along one sliding period. For clarity, a zoom in on an area of $6 \times 6 \text{ nm}^2$ near the bottom edge of GB 1 is presented for the top rigid slider layer (purple bonds) and the PolyGr layer (pink and cyan bonds, representing hexagonal and dislocation atoms). The periodicity ($\sim 3 \text{ nm}$) along the sliding direction is marked by the red arrows.

“2. The abstract is very unclear and vague in certain ways. It should specify that the system studied is for flat graphene sliding on flat graphene layers (i.e. not a tip on graphene for example); that MD is used to obtain the results; that they are studying kinetic, not static friction; and that they are simulating at 5 m/s (to distinguish the work from, e.g., AFM-based studies).”

We thank the reviewer for raising this point and for providing useful suggestions to improve the paper’s abstract, which was revised and now reads as follows:

“The effect of corrugated grain boundaries on the frictional properties of extended planar graphitic contacts incorporating a polycrystalline surface, are investigated via molecular dynamics simulations. The kinetic friction is found to be dominated by shear induced buckling and unbuckling of corrugated grain boundary dislocations, leading to a nonmonotonic behavior of the friction with normal load and temperature. The underlying mechanism involves two effects, where an increase of dislocation buckling probability competes with a decrease of the dissipated energy per buckling event. These effects are well captured by a phenomenological two-state model, that allows for characterizing the tribological properties of any large-scale polycrystalline layered interface, while circumventing the need for demanding atomistic simulations. The resulting negative differential friction coefficients obtained in the high-load regime can reduce the expected linear scaling of grain-boundary friction with surface area and restore structural superlubricity at increasing length-scales.”

We note that we opted not to mention in the abstract the specific value of the sliding velocity employed in our simulations, as the revised manuscript we have demonstrated that the results are independent of this parameter over a wide range of sliding velocities.

“The authors do not actually show that their interfaces are superlubric. The definition of superlubricity is either a very low absolute (not differential) friction coefficient (<0.01), or a very low shear strength. Here, the friction forces are in the range of 0.01-0.05 nN. Using the smaller end of this range, I calculate a shear stress of approximately 30 kPa, which is indeed impressively low (assuming the area is as given in Supp. Fig. 1). Even at 300K and zero stress, friction force is approx. 0.04, or 120 kPa. This should be specified in the manuscript and compared to experimental measurements, which are available.”

We thank the reviewer for raising this important point. Following her/his suggestion, we estimated the friction coefficients at zero temperature, and the upper bound of the frictional stress for 150 K at zero normal load. Furthermore, we compared the results with available experimental measurements. Accordingly, we have made the following changes in the revised manuscript:

On page 3 of the revised main text, the sentence “The corresponding effective friction coefficients obtained at the low and high pressures regimes are 1.5×10^{-4} and -8.1×10^{-5} respectively, well within the superlubric regime.” has been added.

On page 4, the sentence “The differential friction coefficients calculated in this case are between -1.16×10^{-4} and -8.5×10^{-6} .” was revised and augmented as follows: “The differential friction coefficients calculated in this case are between -1×10^{-4} and -8.9×10^{-6} . Compared to experiments, the maximal frictional stress obtained in our simulations (~ 160 kPa) is less than one order magnitude higher than that measured for misaligned homogeneous pristine graphitic contacts^{4,29}, and comparable to that of aligned graphite/hexagonal boron nitride (*h*-BN) heterojunction¹⁰” According to our up-to-date simulation results. Fig. 2a is updated as well with minor differences from its previous version.

On page 13 of revised Supplementary Note 2, we added the following paragraph “Since the friction force has a non-monotonic dependence on the normal load, we estimated the effective friction coefficients via linear fits to the force versus pressure diagrams in the low and high normal load regimes (see Supplementary Fig.6). At zero temperature, the slopes of these linear fits yield friction coefficients of 1.5×10^{-4} and -8.1×10^{-5} , respectively. Here, the friction force values obtained under normal forces of 0 and 627.2 nN were excluded from the fit, as they deviate from the linear segments and correspond to negligible friction forces. For $T = 300$ K, the corresponding friction coefficients are -1×10^{-5} and -8.9×10^{-5} , respectively. Overall, these values are well below the threshold of 10^{-3} for superlubric sliding.”.

On page 14 of the SI, Fig. R9 was added as Supplementary Fig. 6.

Fig. R9. Estimation of the effective friction coefficients. Linear fits (red lines) to the low- and high-load regimes of the friction-force versus normal force curves obtained at (a) $T = 0$ K and (b) $T = 300$ K.

“4. The size of the system is not given in the main text, making it hard to connect forces and stresses. Adding the dimensions to Fig. 1 (as are shown in Supp. Fig. 1) would be helpful.”

We thank the reviewer for pointing out this issue. In the revised manuscript, the dimensions are added to Fig. 1 as suggested.

Minor points of Reviewer #3:

“5. Also, the phrase in the abstract “where an increase of dislocation buckling probability is accompanied by a decrease of the dissipated energy per buckling event” is a bit confusing since “accompanied by” suggests the two effects occur together, when in fact one dominates the other. Perhaps “competes with” would work better”

We thank the reviewer for this valuable suggestion. The revised sentence in the abstract now reads: “The underlying mechanism involves two effects, where an increase of dislocation buckling probability competes with a decrease of the dissipated energy per buckling event.”

“6. Fig 1d would be clearer if the authors labeled the different layers in with their angles (i.e. add a label showing that the top 3 are at 38.2°; the next one is at 0° and 8°, the bottom at 0°). As well, adding the layer number to Fig. 1a and 1d would be helpful. Related to this, the author found it hard to visualize the description in the first two paragraphs under “Results” since the shear plane not specified in Fig. 1. A label or line clearly showing the shear plane should be added to Fig 1a and 1d. Also, the “slider” and the “substrate” should be clearly labeled in Fig. 1a and 1d.”

We thank the reviewer for these helpful suggestions. We have made the corresponding changes in Fig. 1 of the revised manuscript. We have further added the sentence “The shear plane at the interface between layers l_3 and l_4 is denoted by the green line (see also green arrow in panel (a))” to the figure caption.

“7. A movie showing all 5 layers, illustrated in a 3d perspective view, would be extremely helpful. This way, one could see the sliding interface. The movie could start by showing all 5 layers, then repeated with the top 3 layers removed so one can see the buckling.”

We thank the reviewer for this extremely helpful suggestion. We recreated Movie 1 and revised the movie caption in the SI accordingly.

“8. The authors may wish to note that the pentagon-heptagon defects are known also as a type of Stone-Wales defects in the literature.”

We thank the reviewer for this suggestion. We would like to stress that conventional Stone-Wales defects are created by a 90° rotation of a C-C bond with respect to its midpoint, which results in adjacent pairs of pentagon and heptagon rings embedded within a pristine graphene environment. In the case of grain boundaries, individual pentagon-heptagon defects form a seamline between rotated graphitic surface grains. While the two are related, we opted not to use the explicit term of a Stone-Wales defect in order to avoid unnecessary reader confusion.